# A Charging Planning Method for Shared Electric Vehicles with the Collaboration of Mobile and Fixed Facilities

Qingyu Luo , Zhihao Ye and Hongfei Jia *

Transportation College, Jilin University, Changchun 130022, China; luoqy@jlu.edu.cn (Q.L.);
yezh21@mails.jlu.edu.cn (Z.Y.)
*   Correspondence: jiahf@jlu.edu.cn

**Abstract:** Faced with the charging difficulties of free-floating shared electric vehicles and the high cost of single-demand mobile charging, this paper proposes a cooperative charging planning method based on the complementary advantages of fixed charging stations and mobile charging vehicles, which can charge shared electric vehicles more efficiently and reduce the charging cost at the same time. A bi-level programming model for fixed and mobile cooperative charging is constructed. The upper level of the model is the system charging total cost minimization model, which searches for the optimal charging scheme and number of mobile charging vehicles. The lower level model is a fixed and mobile cooperative charging path planning model, which calculates the optimal routes for the mobile charging vehicles and the shared electric vehicles that need to be transferred to the fixed charging station. The example results show that the cost of the proposed fixed-mobile cooperative charging scheme is reduced by 12.6% when compared to the fixed-only charging scheme, and by 14.9% when compared to the mobile-only charging scheme.

**Keywords:** shared electric vehicle; mobile charging vehicle; fixed charging station; cooperative charging



## 1. Introduction

With the development of electric vehicles and related infrastructure [1–3], a new model of electric vehicle charging is beginning to gain traction. In contrast to the fixed charging mode, in this new model we can install a charger on a movable object and then send the charger to serve the electric vehicle (EV) on demand. From the customer's point of view, they can request the charging service as soon as they have parked their EV where they want, which means that the charging activity does not interfere with their schedules. This type of charging mode is called "mobile charging" [4,5]. In practice, some companies have started to offer mobile charging services. Researchers have also achieved some results in the field of mobile charging scheduling. For example, Huang et al. [6] designed a mobile charging service model based on the nearest-work-next rule. Raeesi et al. [7] modelled the mobile charging system as a vehicle routing problem and formulated it with a mixed-integer linear program. Tang et al. [8] established a simulation-based optimization framework to simulate and verify the design and operation of the mobile charging system.

Free-floating shared electric vehicles [9,10] have developed rapidly in recent years and are favored by users due to their convenient rental and unrestricted parking. However, free-floating shared electric vehicles are usually parked in locations without charging stations, and it is too costly to manually move shared electric vehicles (SEVs) to charging stations for charging, and it is difficult to charge SEVs without sufficient power to reach fixed charging stations. Mobile charging services have flexible working characteristics. Mobile charging vehicles (MCVs) can arrive at the request location to provide charging services based on the request information, or they can choose a certain fixed location and charge multiple SEVs in the vicinity of that location. For SEV operators, choosing MCVs to charge at their parking locations solves the challenge of the high cost of charging free-floating SEVs.

At the same time, users of free-floating shared electric vehicles show a significant periodic regularity in their daily journeys, and a cluster distribution in some popular locations. Given that operators need to replenish power for SEVs in a timely manner to ensure that they can serve more users, the introduction of MCVs, which give full play to the advantages of cluster distribution of SEVs and reduce charging costs, become a charging choice that operators can prioritize. Due to the high cost of MCV charging, operators need to further consider how to combine the original fixed charging mode with the new mobile charging mode, as well as rationalize the arrangement of different MCVs to serve different clusters of SEVs, in order to provide charging services to SEVs in a more flexible and cost-effective way.

The three main issues addressed in this paper are:

(1) Can mobile charging services be introduced to solve the problem of the difficult and costly charging of free-floating SEVs?
(2) How can operators plan flexible mobile charging vehicles and low-cost fixed charging stations to achieve the most economical charging scheme?
(3) How can operators efficiently design a travel path of charging SEVs that provides two types of charging with the power distribution and location distribution characteristics of free-floating SEVs?

## 2. Literature Review

Existing studies on free-floating SEVs mainly include the dispatching of free-floating SEVs and the relocation of charging stations. Weikl and Bogenberger [11] developed an intra-regional SEV dispatch model and an inter-regional SEV dispatch model for a free-floating car-sharing system with a mix of conventional and electric vehicles, taking into account the charging of electric vehicles and the refueling of conventional vehicles. They concluded that free-floating car-sharing systems not only allow operators to increase their profits, but also reduce the average idle time at the end of each trip. Kypriadis et al. [12] proposed a model to optimize the charging and repositioning of free-floating SEVs to maximize revenue and orders, where the repositioning of free-floating SEVs is performed by employees. Folkestad et al. [13] studied the optimal repositioning problem for single- and multi-employee free-floating SEVs, to minimize employee walking costs. Roni et al. [14] studied the relationship between the number of charging stations and the idle time of a free-floating SEV. They constructed an integer model to jointly optimize the resetting of charging stations and the allocation of SEVs to charging stations. Based on millions of data points from free-floating combustion engine vehicle rentals, Cocca et al. [15] developed a discrete event tracking driving simulator to optimize the location of SEV charging stations, and based on the study of charging station locations, a data-driven optimization method was proposed to simultaneously optimize the location of charging stations and the number of chargers.

Existing research on EV charging is mainly based on the fixed charging mode, setting charging thresholds, developing charging cycles and performing vehicle scheduling.

Boyaci et al. [16–18] constructed a multi-objective mixed-integer linear programming model to determine the optimal number of vehicles and the optimal location and number of stations for a one-way shared electric vehicle system, but the charging process was not explicitly modelled [19]. Brandstätter et al. [20] investigated the problem of strategic planning levels for a one-way shared vehicle system using electric vehicles to maximize the revenue of the shared vehicle system without exceeding the budget, by identifying the optimal location of charging stations.

Hua [21] proposed an innovative framework for deploying a one-way shared electric vehicle (EV) system serving an urban area. Huang et al. [22] developed a mixed-integer nonlinear programming model to determine the allocation of one-way shared EVs in a given area, which was divided into two parts, decision level and operation level, where the decision level was used to determine the number of vehicles required for operation and the capacity of stations, and the operation level determined the scheduling of vehicles. In a

subsequent study comparing operator-based and user-based vehicle scheduling, Huang et al. [23] used the same continuous distribution to model changes in vehicle performance.

Xu and Meng [24] maximized the operating revenue of shared electric vehicle companies by determining the number of electric vehicles. In determining the number of vehicles to be operated, whether the vehicles are charged or not, the charging time and vehicle scheduling are optimally considered together. Gambella [25] equalized the regional distribution of electric vehicles by using a scheduler to perform scheduling tasks on the vehicles, explicitly considering the consumption of electric vehicle batteries and the charging process.

Research on mobile charging mainly focuses on how to reasonably arrange the scheduling of mobile charging vehicles and path optimization. Qi [26] proposed an optimal path selection method for mobile charging vehicles and solved it using an improved genetic algorithm, in which traffic information such as road congestion, road class and road length were considered in the path selection. When an EV arrives at a station where the chargers are fully occupied, it has to wait for a longer time to charge. Atmaja et al. [27] considered the use of mobile charging vehicles to provide additional charging capacity to overloaded stations and designed a strategy for scheduling mobile charging vehicles in the network. In order to solve the problem of the difficulties of charging electric vehicles, Bao et al. [28] studied the reasonable distribution of mobile charging vehicles as well as scheduling optimization; they first determined the number of mobile charging vehicles needed in the service area, as well as the initial location, and then established a scheduling model of mobile charging vehicles with the goal of satisfying the demands in the shortest possible time. Huang et al. [23] proposed that the use of mobile charging to replenish EVs could compensate for the lack of charging infrastructure coverage and described a strategy for mobile charging to serve the closest users, where each charging vehicle serves the next spatially closest EV after completing its current charging demand. Çalık et al. [29] argued that mobile charging can improve the convenience of charging for EV users, and their paper was the first attempt to propose a bi-level simulation-based model to describe the planning and operational aspects of mobile charging systems.

With the popularity of shared electric vehicles and the rise in mobile charging, the optimization problem of MCVs in mobile charging mode has become a hot research topic. The related research on shared electric vehicles has already achieved more results regarding path optimization and vehicle deployment, but the research on MCV deployment currently has the following shortcomings.

(1) MCV is flexible but costly, while fixed charging is economical and relies on the manual transfer of the SEV to complete charging. A search of the literature shows that there is no published research on how to combine the two charging facilities to achieve the most economical and efficient charging scheme.
(2) Existing studies on MCVs mainly focus on one-to-one charging, and such a charging service has relatively high costs and a low efficiency. The cluster distribution characteristics of free-floating SEVs make it possible to use MCVs for multi-user parallel charging, and research on matching MCVs with SEV clusters needs to be conducted in depth.
(3) On the one hand, there is manual transfer SEV path planning, which is based on the location of fixed charging stations, and, on the other hand, there is MCV path planning, which takes into account SEV clusters. The coordination of fixed and mobile charging planning is worth investigating, based on the power distribution characteristics and location characteristics of SEVs.

## 3. Methodology

### 3.1. Scenario Description

In order to better understand the stationary versus mobile charging methods in this paper, a conceptual illustration is given in Figure 1.

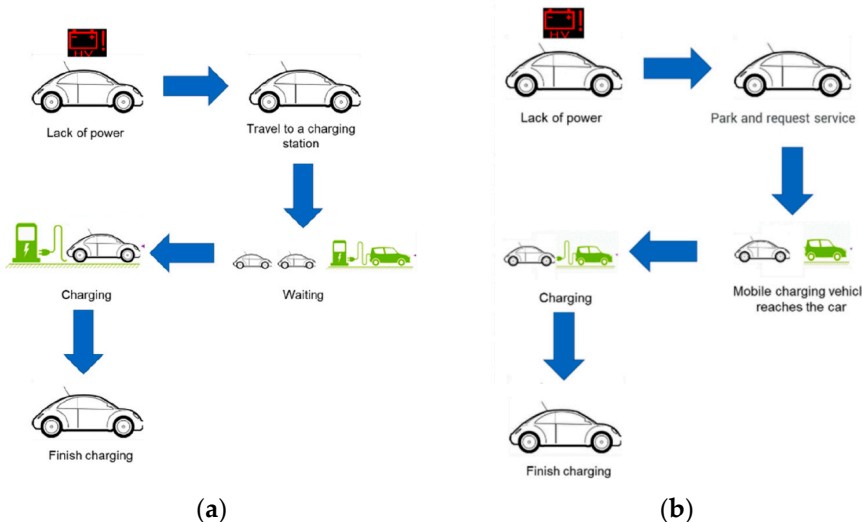

**Figure 1.** Concepts of fixed charging and mobile charging. (**a**) Fixed charging; (**b**) mobile charging.

Assuming that MCVs are purchased by the SEV operator, it is known that the set of SEV cluster nodes are denoted by C = {1, 2,. . .} and the set of charging stations are denoted by F = {1, 2,. . .}, where the numbers in brackets represent cluster node and charging station numbers, respectively, and N-1 denotes the total number of nodes in the SEV cluster. As shown in Figure 2, the dispatched MCVs all start from site 0, travel to the customer node to provide the charging service and return to site N after completing the assigned charging scheme. In fact, MCV depots 0 and N are the same location. Each SEV cluster node can contain several SEVs with different charging needs and idle time windows. One or more SEVs parked at the same SEV cluster node with overlapping time windows can be charged simultaneously or sequentially by an MCV, with the charging routes indicated by black arrows. If the MCV does not have enough energy to provide the charging service or to return to site N, the MCV can be replenished by charging at the site or charging station. In addition, SEVs can be charged by fixed charging with a human driver transferring to a charging station. This type of charging route is indicated by a red arrow.

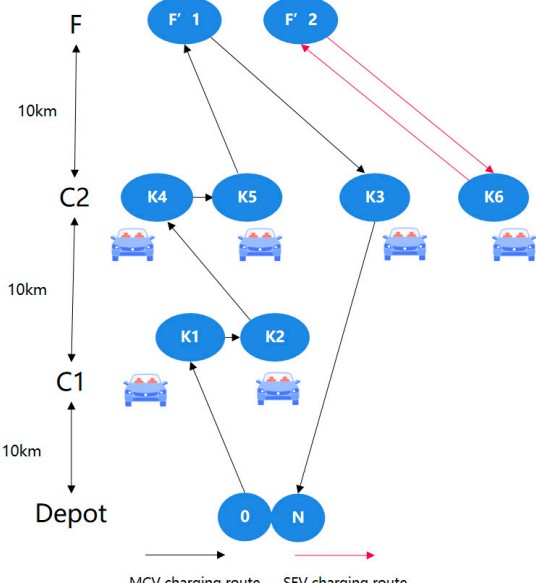

**Figure 2.** Scenario description. Where K1–K6 is the number of the charging vehicle and F′1, F′2 are the number of trips to the stationary charging station in the representative scheme.

Due to the limited battery capacity of MCVs and the limited idle time window of SEVs, the selected SEVs of the same SEV cluster node can be charged simultaneously or sequentially by MCVs. A set K = {1, 2,..., N−1} of virtual SEV cluster nodes is defined, where the numbers represent the virtual SEV numbers. By setting K, SEVs parked at the same SEV cluster node are assigned to different virtual cluster nodes, and the travel distance between virtual SEV cluster nodes, corresponding to the same SEV cluster node, is zero. With this setup, each SEV in the SEV cluster node is set up as a virtual node, so that the SEVs can be numbered independently. Similar to the setting K of the virtual SEV cluster nodes, the setting F' of the virtual charging stations is also defined based on the set F of charging stations, where the distance between the virtual charging station nodes corresponding to the same charging station is zero. Each virtual charging station node in F' can be accessed, at most, once, and the setting of F' makes it easy to calculate the number of times each charging station is accessed in fixed or mobile charging modes. Based on the above scenario, the optimal charging scheme and path optimization are investigated by coordinating mobile and fixed charging.

*3.2. Assumptions and Variables*

In order to simplify the computation of the model, the road environment is assumed to be ideal in this paper, which can reduce the change in relevant parameters due to the change in road environment. The following hypotheses are made in response to the research questions:

- Speed and power consumption rate are constant. The SEV or MCV travel time is equal to the distance between nodes divided by the speed, and the travel power consumed between nodes is equal to the distance between nodes multiplied by the power consumption rate.
- The period of the charging scheme is fixed.
- The charging time of each SEV is fixed.
- The SEV charging threshold is a fixed value. SEVs with power below this threshold send a charging request, and SEVs with power below the requirement to reach the nearest fixed charging station stop their service to wait for MCV charging.

The parameter list is shown in Table 1:

**Table 1.** List of parameters used in this paper.

| Parameter | Define |
|---|---|
| $f_m$ | Number of mobile charging vehicles (veh) |
| $\beta_1$ | Charging price (RMB/kWh) |
| $\beta_2$ | Depreciation cost per MCV (RMB) |
| $\beta_3$ | Initial price of manually transferring SEV to fixed charging station (RMB) |
| $\beta_4$ | Mileage price per kilometer when manually transferring SEVs (RMB/km) |
| $i, j$ | virtual node |
| $x^m_{ijk}$ | $x^m_{ijk} \in \{0, 1\}$, indicates whether the virtual SEV node $k$ is charged by MCV. $x^m_{ijk} = 1$, indicates that SEV $k$ is charged by MCV. |
| $x^s_{ijk}$ | $x^s_{ijk} \in \{0, 1\}$, indicates whether the virtual SEV node $k$ is charged in a fixed mode or not. |
| $h$ | Power consumption per kilometer of MCV (kWh/km) |
| $q^m_k$ | Charging demand for virtual SEV node $k$, which is handled by a mobile charging vehicle(kWh). |
| $q^s_k$ | Charging demand for virtual SEV node $k$, which is covered by a fixed charging station (kWh). |
| $P$ | Battery capacity of SEV (kWh) |
| $p_{ik}$ | Electricity of SEV node $k$ (kWh) |
| $y_i$ | The remaining power when MVC reaches at node $i$ (kWh) |

**Table 1.** *Cont.*

| Parameter | Define |
|-----------|--------|
| $Q$ | Battery capacity of MVC (kWh) |
| $Y_i$ | Charging quantity of MVC at fixed charging stations. $i \in V' \cup \{0\}$ (kWh) |
| $d_{ij}$ | Path distance between nodes $(i, j)$ (km) |
| $\tau_i$ | Time to start services for SEV on the virtual client node |
| $e_i$ | Start time of service time window |
| $l_i$ | End time of service time window |
| $t_{ij}$ | The travel time of a road segment, which is related to the distance of segment $d_{ij}$ and the fixed speed of MCV (min). |
| $t_i$ | The time that a mobile charging vehicle spends on charging a shared electric vehicle at node $i$ (min) |
| M | The length of the longest link in the path set L (km) |
| $s_{ij}$ | $s_{ij} \in \{0,1\}$, indicates whether the charging services at the virtual SEV cluster nodes can be performed at the same time. |

*3.3. The Bi-Level Programming Model for Cooperative Charging*

In order to address the charging scheme issues raised above, a bi-level programming model for fixed and mobile cooperative charging is constructed. The upper model is the total system charging cost minimization model, which searches for the optimal charging scheme and number of mobile charging vehicles. The lower model is a fixed and mobile cooperative charging path planning model, which calculates the optimal routes for the mobile charging vehicles and the shared electric vehicles that need to be transferred to the fixed charging station.

3.3.1. The Upper Model

From the operator's point of view, the upper level of the bi-level programming model focuses on the issue of charging costs, with the aim of deriving the optimal charging scheme and the number of MCVs to be dispatched. The objective of the upper model is to minimize the total system charging cost. The total system charging cost consists of the following two components: the mobile charging cost and the fixed charging cost.

(1) Mobile charging costs;

The cost of mobile charging consists of the following components:

(1) Cost of vehicle use. The purchase cost of the MCV is converted into the depreciation cost for a single trip with reference to the current toll model, as in Formula (1):

$$C_1^m = \beta_2 f_m \tag{1}$$

(2) Time window penalty cost. If the MCV arrives earlier than the time window, it must wait and pay the corresponding early arrival penalty cost $p_e$. If the MCV arrives later than the time window then there is a penalty cost for a late arrival $p_l$. Time window violations on each service path are penalized by calculating only the time window violation on the first virtual SEV cluster node, rather than calculating the idle time window violation on the entire path. The time window penalty is defined in Formula (2):

$$C_2 = \sum_{i \in V} W(t_i)$$
$$W(t_i) = \begin{cases} p_e(e_i - t_i) & , t_i < e_i \\ 0 & , e_i < t_i < l_i \\ p_l(t_i - l_i) & , t_i > l_i \end{cases} \tag{2}$$

(3) Charging cost. This is made up of the MCV's travelling power cost and recharging cost. Travelling power cost is the cost of the power consumed by the MCV during its

journey. The charging cost includes the cost consumed by the MCV at the fixed charging station and the cost consumed to charge the SEV, as in Formulas (3) and (4):

$$C_3^m = \beta_1 \left( \sum_{k \in K_M} d_{ij} x_{ijk}^m h + \sum_{k \in K_M} q_k^m \right) \tag{3}$$

$$q_k^m = P - p_{ik}, \forall i \in K, \forall k \in K_M \tag{4}$$

(2)　Fixed charging cost.

The fixed costs are made up of the following components:

(1) Labor cost. This refers to the cost of manually driving SEVs to the fixed charging station, which consists of the initial cost and the cost per kilometer, as in Formula (5):

$$C_1^s = \sum_{k \in K_S} \beta_3 x_{ijk}^s + \sum_{k \in K_S} d_{ij} x_{ijk}^s \beta_4 \tag{5}$$

(2) Charging cost. This is the cost of SEV charging at a fixed charging station, as in Formulas (6) and (7):

$$C_3^s = \beta_1 \sum_{k \in K_S} q_k^s \tag{6}$$

$$q_k^s = P - p_{ik} + h d_{ij}, \forall i \in K, \forall k \in K_S \tag{7}$$

In summary, the upper level of the bi-level programming model is defined by Formula (8):

$$\min Y = \beta_1 \left( \sum_{k \in K_M} d_{ij} x_{ijk}^m h + \sum_{k \in K_M} q_k^s + \sum_{k \in K_S} q_k^m \right) + \sum_{k \in K_S} d_{ij} x_{ijk}^s \beta_4 + \beta_2 f_m + \sum_{k \in K_S} \beta_3 x_{ijk}^s + \sum_{i \in V} W(t_i) \tag{8}$$

### 3.3.2. The Lower Model

The lower level of the bi-level programming model is a path planning model for cooperative charging by fixed and mobile modes. The purpose is to find the optimal routes for MCVs and SEVs based on the charging scheme from the upper model, minimizing the path planning cost, as in Formula (9):

$$\min Z = \beta_1 \left( \sum_{i \in V' \cup \{0\}, j \in V' \cup \{N\}, i \neq j, k \in K_M} d_{ij} x_{ijk}^m h + \sum_{i \in K, k \in K_s} q_{ik}^s + \sum_{i \in K, k \in K_M} q_{ik}^m \right) + \sum_{i \in V', j \in F', i \neq j, k \in K_s} d_{ij} x_{ijk}^s \beta_4 + \sum_{i \in V} W(t_i) \tag{9}$$

There are some constraints of the lower model, as follows:

(1)　Charge mode constraint;

By classifying SEVs below the charge threshold into either fixed or mobile charging modes, Formula (10) ensures that all SEVs below the charge threshold are charged using one of these methods:

$$\sum_{j \in V' \cup \{N\}, i \neq j, k \in K_M} x_{ijk} + \sum_{j \in V' \cup \{N\}, i \neq j, k \in K_S} x_{ijk} = 1, \forall i \in K \tag{10}$$

(2)　Vehicle equilibrium constraint;

$$\sum_{j \in V' \cup \{N\}, i \neq j, k \in K} x_{jik} = \sum_{j \in V' \cup \{0\}, i \neq j, k \in K} x_{ijk}, \forall i \in V' \tag{11}$$

Formula (11) shows that the total number of vehicles in any two nodes, except for the depot, remains constant, ensuring that the total number of MCV and SEV vehicles does not change.

(3) Time window constraint;

$$e_i \leqslant \tau_i \leqslant l_i, \forall i \in V' \cup \{0, N\} \tag{12}$$

Formula (12) ensures that the charging services provided by the MCVs are counted from the beginning of the time window.

(4) Fixed charging station constraint;

$$\sum_{j \in V' \cup \{N\}, i \neq j} x_{ij} \leqslant 1, \forall i \in F' \tag{13}$$

Since each visit to a charging station node generates a new virtual charging station node, Formula (13) is proposed, to ensure that each virtual charging station node is visited, at most, once.

$$p_{ik} - hd_{ij}x_{ik} + P\left(1 - x_{ijk}^s\right) \geq 0, \forall i \in V', \forall j \in F', i \neq j, \forall k \in K_S \tag{14}$$

Formula (14) ensures that the power of the SEV, when leaving the virtual node, is sufficient to reach the fixed charging station.

$$\tau_i + t_{ij}x_{ijk}^s + \alpha s_{ij} - l_0\left(1 - x_{ijk}^s\right) \leqslant \tau_j, \forall i \in K, \forall j \in V', i \neq j, k \in K_S \tag{15}$$

Formula (15) ensures the temporal feasibility of SEVs leaving the links of SEV virtual cluster nodes and virtual charging station nodes.

$$p_{ik} \leq hd_{ij}, \forall i \in K, \forall j \in F', k \in K_M \tag{16}$$

The current power level of the electric vehicle is judged, and it is classified into the mobile charging scheme when the power is not enough to reach the fixed charging station.

(5) Related constraints of mobile charging mode.

$$x_{ijk}^m q_j \leqslant y_j \leqslant y_i - hd_{ij}x_{ijk}^m - q_i x_{ijk}^m + Q\left(1 - x_{ijk}^m\right), \forall i \in K, \forall j \in V' \cup \{N\}, i \neq j, k \in K_M \tag{17}$$

$$x_{ijk}^m q_j \leqslant y_j \leqslant Y_i - hd_{ij}x_{ijk}^m + Q\left(1 - x_{ijk}^m\right), \forall i \in F' \cup \{0\}, \forall j \in V' \cup \{N\}, i \neq j, k \in K_M \tag{18}$$

$$y_i \leqslant Y_i \leqslant Q, \forall i \in F' \cup \{0\} \tag{19}$$

Formulas (17) and (18) ensure that the power state of the MCVs, when leaving the virtual customer node, the virtual charging node and the starting point, satisfies the subsequent services, respectively, and Formula (19) places a limit on the MCVs' power. Formula (18) is the MCV's power relationship between two nodes, in the case of $x_{ijk}^m = 1$. The left-hand side constraint ensures that the remaining power of the MVC to reach node $j$ is not less than the charging demand of node $j$, and the right-hand side constraint ensures that the remaining power of the MVC $y_j$ to reach node $j$ is greater than the sum of the power demand and trip consumption at the two nodes of $(i, j)$.

$$\tau_0 + t_{0j}x_{0jk}^m - l_0\left(1 - x_{0j}^m\right) \leqslant \tau_j, \forall j \in V' \cup \{N\}, k \in K_M \tag{20}$$

$$\tau_i + t_{ij}x_{ijk}^m + \alpha t_i - l_0\left(1 - x_{ijk}^m\right) \leqslant \tau_j, \forall i \in K, \forall j \in V' \cup \{N\}, i \neq j, k \in K_M \tag{21}$$

$$\tau_i + t_{ij}x_{ijk}^m + g(Y_i - y_i) - (l_0 + gQ)\left(1 - x_{ijk}^m\right) \leqslant \tau_j, \forall i \in F', \forall j \in V' \cup \{N\}, i \neq j, k \in K_M \tag{22}$$

Formulas (20)–(22) ensure the temporal feasibility of the linking of MCVs leaving depot 0, SEV virtual cluster nodes, and virtual charging station nodes, respectively.

In order to ensure that the MCV can simultaneously charge multiple SEVs at the same time without exceeding the limit on the number of simultaneous charges, the parallel mobile charging constraint is proposed, as follows:.

$$\frac{d_{ij}x_{ijk}^m}{M} \leqslant s_{ij} \leqslant Kd_{ij}x_{ijk}^m + \frac{d_{ij}}{M}, \forall i \in K, \forall j \in V' \cup \{N\}, i \neq j, k \in K_M \tag{23}$$

$$\min\left\{x_{ijk}^m, 1 - s_{ij}, \max\left\{\frac{\tau_i + \alpha - \tau_j}{E}, 0\right\}\right\} \leqslant R_{ij}, \forall i \in V' \cup \{0\}, \forall j \in V' \cup \{N\}, i \neq j, k \in K_M \tag{24}$$

$$\Phi_j = \max\{(\Phi_i + R_{ij})R_{ij}, 1\}, \forall i \in V' \cup \{0\}, \forall j \in C', i \neq j \tag{25}$$

$$0 \leqslant \Phi_i \leqslant H, \forall i \in V' \cup \{0, N\} \tag{26}$$

Formula (23) ensures that the services of two nearby nodes can be provided simultaneously or sequentially by the same MCV, while the charging services of two more distant virtual nodes can only be provided sequentially. Formula (24) is used to determine whether the two SEVs located at virtual nodes are simultaneously charged by the same MCV. Formula (25) determines the number of SEVs simultaneously charged by the MCVs. Formula (26) ensures that the number of SEVs simultaneously charged by one MCV does not exceed the number limit.

### 3.4. GASA Algorithm

Since the solved fixed–mobile cooperative path planning problems are NP-hard problems [30,31], solving such problems using exact algorithms takes a long time and it is difficult to obtain a global optimal solution. Liu [32] combined the advantages of a genetic algorithm and a simulated annealing algorithm and proposed a genetic–simulated annealing algorithm (GASA) to solve the bi-level programming model; GASA has a stronger search ability and efficiency, and it can obtain a global optimal solution.

For the given fixed and mobile cooperative charging bi-level programming model, the entire bi-level programming model is solved using the GASA algorithm, and the lower level of the bi-level programming model is solved using the genetic algorithm. The solution ideas are as follows:

Step 1. Set the genetic parameters: population size $N$, crossover probability $P_c$, mutation probability $P_m$, and evolutionary generation $g$. Set the simulated annealing parameters and perform binary coding on the feasible solutions;

Step 2. Given an initial charging scheme population $\left(x_{ijk}^{m(g)}, x_{ijk}^{s(g)}\right)_{N'}$ $g = 0$;

Step 3. Calculate the fitness function values (the upper level of the Bi-level programming model objective function value) for the individuals in the population, as described below:

(1) Set the initial solution under the individual charging scheme with iteration number n = 0. The initial solution is obtained by continuously constructing the feasible route;

(2) Calculate the broad charging costs $Y$;

(3) Use genetic algorithms to find optimal populations $\left(x_{ijk}^{m(g)}, x_{ijk}^{s(g)}, f_m^{(g)}\right)$;

(4) Substitute the output population $\left(x_{ijk}^{m(g)}, x_{ijk}^{s(g)}, f_m^{(g)}\right)$ into the upper model to find the value of the fitness function.

Step 4. Perform genetic operations such as replication, crossover and mutation on the population, based on the value of the fitness function to obtain a new charging scheme population $\left(x_{ijk}^{m(g+1)'}, x_{ijk}^{s(g+1)'}, f_m^{(g+1)'}\right)_{N'}$;

Step 5. Perform a simulated annealing operation on the individuals in the population to obtain the charging scheme population $\left( x_{ijk}^{m(g+1)}, x_{ijk}^{s(g+1)}, f_m^{(g+1)} \right)_N$;

Step 6. Execute Step 3 for the individuals in the population of charging scheme $\left( x_{ijk}^{m(g)}, x_{ijk}^{s(g)}, f_m^{(g)} \right)_N$ to obtain the value of fitness function and determine whether it is equal to the maximum number of evolutionary generations. Otherwise, set $g = g + 1$, and transfer to Step 4 to continue solving. If it is, output the result, and the individual with the largest fitness in the population is the optimal solution.

## 4. Numerical Example

From the perspective of the SEV operator, MCVs are purchased by the SEV operator. Based on the known information about parking points, idle time windows and the charging demand of SEVs, the key issue is to develop a fixed and mobile cooperative charging scheme and arrange the number of MCVs based on the layout of SEV charging clusters and fixed charging stations within a charging cycle, so as to efficiently charge SEVs below the power threshold and reduce charging costs.

### 4.1. Data Source

The study is based on the operational data of a large-scale shared electric vehicle enterprise in Nanjing, China, and the data used mainly include order data, vehicle data and station location data. A region in Nanjing was selected, and some vehicle information in the corresponding region was extracted from the data, including location, parking time, and power level. The time window was set according to the parking time, and the shared electric vehicles whose electricity was lower than the charging electricity threshold were selected as the objects of example analysis.

To simplify the problem, the involved shared electric vehicle parking points were set as virtual nodes, with the same location as the original nodes. Using the charging station points, the coordinate information, time window, service time and electricity of the shared cars are shown in Tables 2 and 3, where 0 is the MCV station and 51–54 are the fixed charging stations.

**Table 2.** Information about SEVs.

| SEV (id) | Latitude | Longitude | Time Window (min) | | Electricity (%) |
|:---:|:---:|:---:|:---:|:---:|:---:|
| 1 | 118.385968 | 32.705249 | 211 | 247 | 0.32 |
| 2 | 118.170446 | 32.688855 | 50 | 90 | 0.27 |
| 3 | 118.238767 | 32.952442 | 59 | 79 | 0.18 |
| 4 | 118.003842 | 32.81767 | 103 | 133 | 0.23 |
| 5 | 118.093071 | 32.641006 | 35 | 61 | 0.34 |
| 6 | 118.057605 | 32.860424 | 190 | 208 | 0.20 |
| ...... | ...... | ...... | ...... | ...... | ...... |
| 39 | 118.235922 | 32.8217 | 30 | 160 | 0.31 |
| 40 | 118.281175 | 32.938024 | 35 | 150 | 0.35 |
| 41 | 118.346757 | 32.704077 | 20 | 90 | 0.08 |

**Table 3.** Information about fixed charging stations.

| SEV (id) | Latitude | Longitude | Time Window (min) | |
|:---:|:---:|:---:|:---:|:---:|
| 0 | 118.374992 | 32.674618 | 0 | 300 |
| 51 | 118.015484 | 32.871561 | 0 | 300 |
| 52 | 118.19667 | 32.618216 | 0 | 300 |
| 53 | 118.217592 | 32.871277 | 0 | 300 |
| 54 | 118.281175 | 32.938024 | 0 | 300 |

### 4.2. Parameter Setting

The charging scheme cycle was set to 5 h, the charging threshold was set to 40%, and the charging service time per SEV for MCVs was set to be fixed and equal to 20 min, whereas the fixed charging station had a charging time of 60 min for the SEVs, and the charging time for the MCVs was determined by the subsequent charging demand and mileage traveled. Assuming a battery capacity of 50 kWh for SEVs and 800 kWh for MCVs, both traveling at a speed of 30 km/h, and having a consumption rate of 0.2 kWh/km during the trip, the fixed charging station charged a rate of 1 RMB/kWh and the MCV charging price at MCV stations was 0.5 RMB/kWh. For MCVs, the initial price was 100 RMB; for fixed charging mode, the initial price was 10 RMB; and the mileage price was 0.8 RMB/km. Parameters in the penalty function were $p_e = p_l = 300$ RMB/h.

### 4.3. Calculation Result

Using the Matlab programming software R2022a, the problem was solved to obtain the charging scheme, as shown in Table 4, which consisted of the vehicle charging mode division, the number of MCVs, and the driving path. The charging mode division means that each charging SEV will be divided into fixed or mobile mode for charging. The number of MCVs refers to the optimal number of MCVs dispatched in order to ensure that charging demand is met and charging cost is reduced. The optimal charging path is shown in Figure 3, and the traveling path includes the charging path of MCVs and SEVs to fixed charging stations for charging and return paths.

**Table 4.** Calculation results.

|  | **Mobile Charging Mode** | **Fixed Charging Mode** |
|---|---|---|
| SEV (id) | 1, 2, 3, 4, 7, 9, 12, 13, 14, 15, 16, 17, 18, 19, 20, 21, 22, 23, 24, 25, 26, 29, 31, 32, 34, 35, 37, 38, 39, 40, 41 | 5, 6, 8, 10, 11, 27, 28, 30, 33, 36 |
| Number of vehicles | 31 | 10 |
| Charging Path | MCV1:0-(4,31)-(20,21)-16-26-(19,32)-17-(13,14)-(7,34)-(40,41)-0 MCV2:0-25-(1,2,3,15,37)-9-(12,29,38)-(18,39)-35-(22,23,24)-0 | 5-51-5, 6-51-6, 8-51-8, 10-52-10, 11-52-11, 27-52-27, 28-53-28, 30-54-30, 36-55-36 |
| Total cost of charging (RMB) | 2094.8 | |

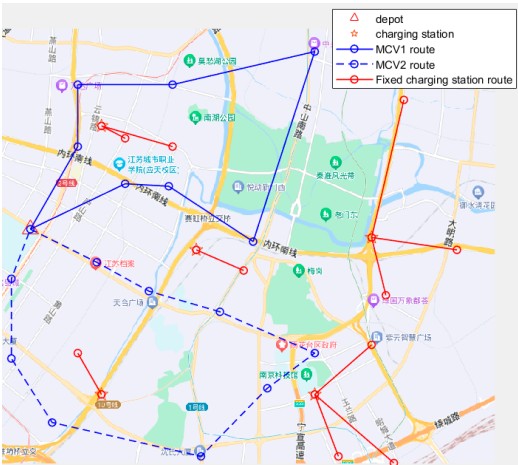

**Figure 3.** Path planning results for fixed and mobile co-charging, exemplified by a region in Nanjing.

*4.4. Analysis of Results*

4.4.1. The Impact of Different Charging Modes on the Cost of Charging Schemes

In order to compare the charging schemes with different charging modes and to verify the practicality of the cooperative charging model, the proposed model is compared with the conventional charging model. In the fixed charging scheme, the SEVs that have enough power to reach the nearest fixed charging station are transferred to the station for charging by people, while the SEVs that do not have enough power to reach the fixed charging station are classified into the mobile charging scheme and charged by the MCVs. While in the mobile charging scheme, all SEVs rely on MCVs for charging, taking into account the constraints of parallel mobile charging. The resulting calculation is shown in Figure 4.

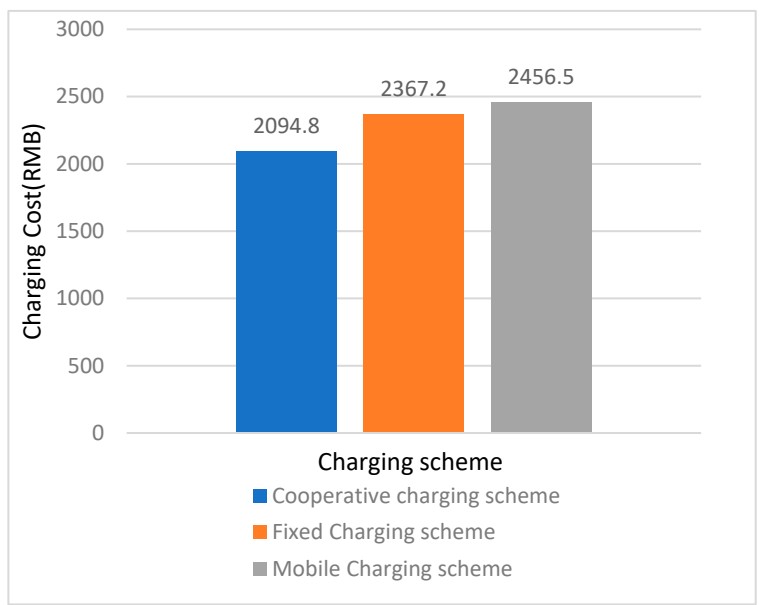

**Figure 4.** The comparison of charging costs by charging scheme.

As can be seen in Figure 4, the proposed fixed and mobile cooperative charging scheme has the smallest charging cost among the three schemes, which is reduced by 12.6% compared to the fixed charging scheme and 14.9% compared to the mobile charging scheme. The results demonstrate the effectiveness of the planning methodology. In this example, the total cost of the mobile charging scheme is 4.1% higher than the fixed charging scheme, mainly because MCV depots are generally not located in the city center. In order to meet all the charging requirements within the charging cycle and avoid the time window penalties, more MCVs would be needed and used to charge SEVs, and charging the SEVs that are too far away would increase the charging costs significantly.

4.4.2. Sensitivity Analysis

In order to illustrate the applicability and robustness of the proposed bi-level programming model under the influence of different factors, the charging price, the MCV charging performance, and the time window penalty cost are selected for sensitivity analysis.

(1)    Charging Price

A sensitivity analysis is performed for the three charging schemes to changes in the charging price. It can be seen in Figure 5 that the costs of the three schemes increase with the charging price, with the lowest total charging costs for the fixed and mobile cooperative charging scheme. When the charging price is low, the total cost of the mobile-only charging scheme is higher than that of the fixed charging scheme.

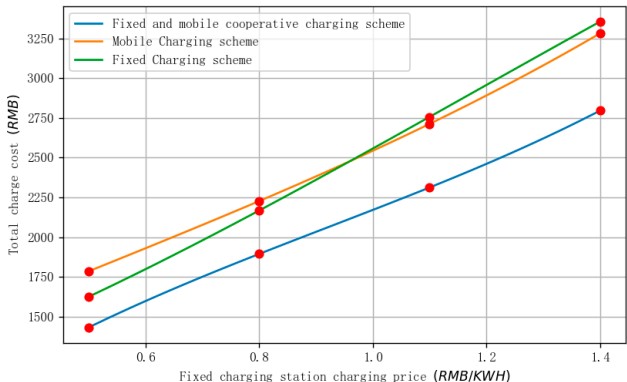

**Figure 5.** The impact of charging price. The red dots here represent the total cost of charging when the fixed charging station charging price is 0.5, 0.8, 1.1, and 1.4 RMB/kWh respectively.

However, when the price increases to a certain level, the fixed charging scheme consumes more electricity than the mobile charging scheme, because the fixed charging system requires each SEV to be dispatched for charging. In this case, the electricity consumed for the kilometers travelled in the fixed charging scheme is much higher than in the mobile charging scheme, and the electricity consumed for the kilometers travelled in the fixed charging scheme increases rapidly with the charging price, resulting in the total cost of the fixed charging scheme being higher than that of the mobile charging system.

(2) MCV Charging Performance

In the fixed and mobile cooperative charging mode, the charging schemes are calculated with different MCV charging capacities and service time, and the results are shown in Tables 5 and 6.

**Table 5.** The impact of MCV charging capacity.

| MCV Battery Capacity (kWh) | Depreciation Cost (RMB) | Number of MCVs | Number of SEVs in Mobile Charging | Number of SEVs in Fixed Charging | Total System Cost (RMB) |
|---|---|---|---|---|---|
| 600 | 80 | 3 | 37 | 4 | 2155.7 |
| 800 | 100 | 2 | 31 | 10 | 2094.8 |
| 1000 | 130 | 2 | 32 | 9 | 2046.5 |
| 1200 | 160 | 2 | 32 | 9 | 2067.3 |

**Table 6.** The impact of MCV charging service time.

| MCV Charging Service Time (min) | Depreciation Cost (RMB) | Number of MCVs | Number of SEVs in Mobile Charging | Number of SEVs in Fixed Charging | Total System Cost (RMB) |
|---|---|---|---|---|---|
| 10 min | 150 | 2 | 35 | 10 | 2056.3 |
| 20 min | 100 | 2 | 24 | 13 | 2094.8 |
| 30 min | 80 | 3 | 31 | 6 | 2227.5 |

As can be seen from Table 5, the total charging cost is related to the battery capacity of the MCV. When the battery capacity of the MCV reaches a certain level, no more vehicles can be served to limit the time window. In addition, the total charging cost increases due to the increasingly depreciating cost of the MCV. Table 6 shows that the total cost of charging is positively correlated with the MCV charging time. If the charging efficiency is improved, the charging cost will decrease. Therefore, SEV companies need to select MCVs with appropriate charging capacity in operation, or even use a combination of MCVs with different charging capacity to reduce the charging cost.

(3) Time Window Penalty Cost

The charging schemes are calculated with different time window penalty costs and the results are shown in Table 7.

**Table 7.** The impact of time window penalty cost.

| Time Window Penalty Cost (RMB/h) | Number of MCVs | Number of SEVs in Mobile Charging | Number of SEVs in Fixed Charging | Total System Cost (RMB) |
|---|---|---|---|---|
| 150 | 2 | 30 | 7 | 1976.3 |
| 300 | 2 | 24 | 13 | 2094.8 |
| 600 | 3 | 31 | 6 | 2323.5 |

From Table 7, it can be seen that, as the cost of the penalty increases, the operator will reduce the number of SEVs served by MCVs and allocate more SEVs to fixed charging stations for charging, in order to reduce the impact of the time window penalty. If the cost of the time window penalty is high to a certain extent, the operator will include more MCVs in the charging program, to ensure that the charging demand can be met without incurring a penalty.

## 5. Conclusions

### 5.1. Contribution

In order to solve the charging difficulties caused by free-floating SEVs parked at will and the high cost caused by the individual charging demands of MCVs, based on the complementary advantages of fixed charging and mobile charging, a bi-level programming model of fixed and mobile cooperative charging is proposed, and the best charging scheme and the number of required MCVS are obtained. Compared with the traditional charging path planning model, the proposed model in this paper has the following innovations:

(1) MCVs serving the single charging demand will lead to a high charging cost. In this paper, MCVs are introduced to serve multiple SEVs, and the parallel mobile charging service is used in the free–mobile SEV charging, which expands the service scope and reduces the charging cost of MCVs.

(2) Utilizing the complementary advantages of fixed charging mode and MCVs, we propose a bi-level programming model for fixed and mobile cooperative charging and solve the model, using the GASA algorithm, to derive the optimal charging scheme and the number of required MCVs, which solves the problems of charging difficulties for free-floating SEVs.

(3) To solve the charging path problem of MCVs and SEVs, a fixed–mobile cooperative charging path planning model and related constraints are proposed, to compute the optimal paths for MCVs and SEVs to be transferred to a fixed charging station for charging.

### 5.2. Discussion

The research in this paper helps free-floating shared electric vehicle companies to reduce their charging costs, but the ways in which to integrate this research with shared electric vehicle charging scheduling will be a major challenge for future research.

In addition, some results have been achieved in the study of path planning problems for shared electric vehicles and mobile charging vehicles, but there are still some shortcomings that need to be investigated in the following papers.

(1) In practice, fixed charging stations often face the problem of charging queues, which increases the cost of fixed charging stations. This article does not consider the queuing problem at fixed charging stations, and the impact caused by queuing should be considered in subsequent studies.

(2)     The road network and setup during driving are ideal. Shared electric vehicle networks of different sizes are not considered, nor are the energy consumption and environmental impacts of charging vehicles. These factors on the SEV charging scheme need to be considered and solved in future studies.

(3)     In subsequent studies, SEV networks of different sizes or characteristics can be analyzed to examine the environment in which the proposed methodology is applicable. The subsequent scheduling issues, as well as the impact of factors such as operating costs and energy expenditure in different regions on scheme planning, need to be investigated.

**Author Contributions:** Conceptualization, Q.L. and Z.Y.; methodology, Z.Y. and Q.L.; writing—original draft preparation, Z.Y.; writing—review and editing, H.J.; funding acquisition, Q.L. and H.J. All authors have read and agreed to the published version of the manuscript.

**Funding:** This research is funded by the Humanities and Social Science Fund of Ministry of Education of People's Republic of China [grant number 20YJCZH115] and the Scientific Research Project of Education Department of Jilin Province (JJKH20231188KJ).

**Institutional Review Board Statement:** Not applicable.

**Informed Consent Statement:** Not applicable.

**Data Availability Statement:** The data presented in this study are available on request from the corresponding author. The data are not publicly available due to it relates to internal data of shared electric vehicle companies.

**Conflicts of Interest:** The authors declare no conflict of interest.

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
