# Peer review of "A Charging Planning Method for Shared Electric Vehicles with the Collaboration of Mobile and Fixed Facilities"

_sustainability, doi:10.3390/su152216107_

Round 1

Reviewer 1 Report

Comments and Suggestions for Authors

Please see the attached Word documents with my comments.

It is a good paper that is well written. There are no fundamental problems with the draft paper, but there is a little cleaning up that needs to be undertaken before it is ready for publication.

My one other comment or suggestion relates to the relevance of the research to countries other than China. In most other countries, mobile charging is a much higher cost option. I think that it would be worth noting that while the results or findings pertain to China, they may not be as applicable in other countries where the costs of mobile EV charging is considerably higher.

Comments on the Quality of English Language

There are a few minor changes required and I have made suggested changes in the Word document attached with my comments

Author Response

Dear Reviewer,

Thank you for your comments concerning our manuscript entitled “A charging planning method for shared electric vehicles with the collaboration of mobile and fixed facilities” (Manuscript ID: sustainability-2699150). These comments are all valuable and very helpful for revising and improving our manuscript, as well as the important guiding significance to our researches. We have studied comments carefully and have made correction which we hope to meet with approval. Revisions are marked in red, such that any changes can be easily viewed. The point-by-point response to your comments is listed as follows.

Question 1: It is a good paper that is well written. There are no fundamental problems with the draft paper, but there is a little cleaning up that needs to be undertaken before it is ready for publication.

Response: We thank you for this helpful suggestion. We have made corresponding changes to each of your suggested changes, including problems with units, grammatical problems, problems in the literature, and a new literature review.

Question 2: My one other comment or suggestion relates to the relevance of the research to countries other than China. In most other countries, mobile charging is a much higher cost option. I think that it would be worth noting that while the results or findings pertain to China, they may not be as applicable in other countries where the costs of mobile EV charging is considerably higher.

Response: Thanks for the constructive comments. The model given in this paper will give different scenario results depending on different parameter settings, which will lead to variability in different regions. In subsequent studies, shared EV networks of different sizes in different regions can be analyzed to examine the environment in which the proposed methodology is applicable and its subsequent scheduling issues, as well as to study the impact of factors such as operating costs and energy expenditures in different regions on the results of the program. (p.15, line 482-486)

We sincerely hope that this revised manuscript has addressed all your comments. We appreciate for your warm work earnestly, and hope that the correction will meet with approval. Once again, thank you very much for your comments and suggestions.

Reviewer 2 Report

Comments and Suggestions for Authors

This article proposes a collaborative charging planning method based on the complementary advantages of fixed charging stations and mobile charging vehicles, aiming to efficiently charge shared electric vehicles while reducing charging costs. The solution presented in this paper results in cost reductions compared to both pure fixed and pure mobile charging schemes. This study offers an effective approach for charging shared vehicles in the field.

The topic of the manuscript is interesting. However, I believe that the article could benefit from some major revisions. The detailed suggestions are listed below:

1. Elaborate on the applicability and robustness of the proposed bi-level programming model in different scenarios.

2. The authors did not consider the energy consumption of charging vehicles and environmental impact factors. It is recommended that the authors address these factors and propose solutions.

3. Was the consideration of different scales of shared electric vehicle networks taken into account? Is the proposed method still applicable for networks of different scales?

4. Can the specific influencing factors behind the mentioned cost reduction ratio, such as operational costs and energy expenses, be further analyzed?

5. Did the authors consider suggesting further research directions, such as exploring how to further optimize the shared electric vehicle charging system, including integration with renewable energy sources?

6. The literature review should be improved significantly, and the latest references need to be supplemented. The following references are recommended for consideration: [1] An optimal charging scheduling model and algorithm for electric buses; [2] Sustainable urban mobility: Flexible bus service network design in the post-pandemic era.

7. It is recommended to conduct a grammar and spelling check on the entire manuscript to ensure the language quality and adherence to academic standards.

Comments on the Quality of English Language

The language should be improved greatly.

Author Response

Dear Reviewer,

Thank you for your comments concerning our manuscript entitled “A charging planning method for shared electric vehicles with the collaboration of mobile and fixed facilities” (Manuscript ID: sustainability-2699150). These comments are all valuable and very helpful for revising and improving our manuscript, as well as the important guiding significance to our researches. We have studied comments carefully and have made correction which we hope to meet with approval. Revisions are marked in red, such that any changes can be easily viewed. The point-by-point response to your comments is listed as follows. 

Question 1: Elaborate on the applicability and robustness of the proposed bi-level programming model in different scenarios.

Response: We thank you for this helpful suggestion. To illustrate the applicability and robustness of the proposed bi-level programming model under the influence of different factors, a sensitivity analysis of some of the influencing factors is done(p.13).

Question 2: The authors did not consider the energy consumption of charging vehicles and environmental impact factors. It is recommended that the authors address these factors and propose solutions.

Response: Thanks for the constructive comments. We have focused the question into future research directions.(p.16, line 478-481) The road network and the setup of the driving process in the paper are idealized. The energy consumption and environmental impact factors of charging vehicles will be considered and solutions will be proposed in future studies.

Question 3: Was the consideration of different scales of shared electric vehicle networks taken into account? Is the proposed method still applicable for networks of different scales?

Response: Thanks for the constructive comments. We have focused the question into future research directions. In subsequent studies, shared electric vehicle networks of different sizes and in different regions will be analyzed to study the context in which the proposed methodology is applicable.(p.16, line 482-486)

Question 4: Can the specific influencing factors behind the mentioned cost reduction ratio, such as operational costs and energy expenses, be further analyzed?

Response: Thanks for the constructive comments. We have focused the question into future research directions.(p.16, line 482-486) In subsequent studies, shared EV networks of different sizes in different regions can be analyzed to examine the environment in which the proposed methodology is applicable and its subsequent scheduling issues, as well as to study the impact of factors such as operating costs and energy expenditures in different regions on the results of the program.

Question 5: Did the authors consider suggesting further research directions, such as exploring how to further optimize the shared electric vehicle charging system, including integration with renewable energy sources?

Response: Thanks for your reminder. We have divided the fifth section into two sub-summaries, (p.15, 5.1contribution and 5.2discussion) in which we present relevant directions for future research in light of some of the issues that have arisen in this paper.

Question 6: The literature review should be improved significantly, and the latest references need to be supplemented. The following references are recommended for consideration: [1] An optimal charging scheduling model and algorithm for electric buses; [2] Sustainable urban mobility: Flexible bus service network design in the post-pandemic era.

Response: We feel sorry for our carelessness. We have completely improved the literature review, supplemented it with up-to-date references, and added relevant literature that you have suggested.

Question 7: It is recommended to conduct a grammar and spelling check on the entire manuscript to ensure the language quality and adherence to academic standards.

Response: We feel sorry for our carelessness. We have checked and revised the full text, including issues of units, grammar, documentation, and so on, to ensure language quality and compliance with academic standards.

We sincerely hope that this revised manuscript has addressed all your comments. We appreciate for your warm work earnestly, and hope that the correction will meet with approval. Once again, thank you very much for your comments and suggestions.

Reviewer 3 Report

Comments and Suggestions for Authors

Dear Authors, this paper investigates the charging planning method for shared electric vehicles. I want to express my appreciation for the effort you have put into your research. However, there are some comments and suggestions to improve the quality of the manuscript. The comments are as follows:

·         The introduction provides a clear context about the challenges of charging free-floating shared electric vehicles and the potential of mobile charging. However, the transition between the general context of electric vehicles and the specific challenges of shared electric vehicles could be smoother.

·         Bullet points or a numbered lists of the main problems proposed in the paper might aid readability.

·         It might be beneficial to provide a rationale for each assumption made in the study, explaining why it's necessary or how it simplifies the model.

·         The literature review could benefit from a more structured presentation, possibly organizing the discussed works chronologically or based on the specific challenges they address. Furthermore, it could be strengthened by offering a more in-depth discussion of the findings of the cited works, a comparative analysis of different methodologies, and a clearer organization of the discussed works. Suggested literature for shared electric vehicles:

o   https://doi.org/10.1016/j.trd.2019.102210

o   https://doi.org/10.1016/j.trb.2019.07.005

o   https://doi.org/10.1016/j.energy.2022.123400

o   https://doi.org/10.1109/TSG.2020.3025082

·         The paper presents results based on the operation data of a large-scale shared electric vehicle enterprise in Nanjing, China. While the data sources are well-defined, the presentation of results could benefit from a more structured format, possibly using subheadings for different aspects of the results. Furthermore, a more detailed discussion on the differences in results, advantages, and limitations of each model might provide more depth to the analysis.

·         The conclusions could benefit from a discussion on the broader implications of these findings for the shared electric vehicle industry or potential challenges in implementing the proposed strategy.

·         It would be beneficial if the conclusions touch upon areas that need further exploration or specific directions for future research.

Author Response

Dear Reviewer,

Thank you for your comments concerning our manuscript entitled “A charging planning method for shared electric vehicles with the collaboration of mobile and fixed facilities” (Manuscript ID: sustainability-2699150). These comments are all valuable and very helpful for revising and improving our manuscript, as well as the important guiding significance to our researches. We have studied comments carefully and have made correction which we hope to meet with approval. Revisions are marked in red, such that any changes can be easily viewed. The point-by-point response to your comments is listed as follows.

Question 1: It might be beneficial to provide a rationale for each assumption made in the study, explaining why it's necessary or how it simplifies the model.

Response: We thank you for this helpful suggestion. We provide an overall explanation of the hypotheses presented for the reader's understanding.(p.5, line 192-194)

Question 2: The literature review could benefit from a more structured presentation, possibly organizing the discussed works chronologically or based on the specific challenges they address. Furthermore, it could be strengthened by offering a more in-depth discussion of the findings of the cited works, a comparative analysis of different methodologies, and a clearer organization of the discussed works. Suggested literature for shared electric vehicles.

Response: Thanks for the constructive comments. We have completely improved the literature review, supplemented it with up-to-date references, and added relevant literature that you have suggested. The literature review concludes with a more in-depth discussion of the findings of the cited literature, raising several issues that arise from the current state of research.(p.3, line 138-151)

Question 3: The conclusions could benefit from a discussion on the broader implications of these findings for the shared electric vehicle industry or potential challenges in implementing the proposed strategy.

Response: We thank you for this helpful suggestion. As suggested, we have divided the fifth section into two sub-summaries, (p.15, 5.1contribution and 5.2discussion). We provide a discussion of the broader implications of these findings for the shared electric vehicle industry or the potential challenges of implementing the proposed strategy. (p.15, line 466-468)

Question 4: It would be beneficial if the conclusions touch upon areas that need further exploration or specific directions for future research.

Response: Thanks for your reminder. We have divided the fifth section into two sub-summaries, (p.15, 5.1contribution and 5.2discussion) in which we propose relevant directions for future research based on some of the issues that have arisen in this paper, including directions such as vehicle scheduling after charging, environmental changes that may be encountered during charging, and so on.

We sincerely hope that this revised manuscript has addressed all your comments and suggestions. We appreciated for your warm work earnestly, and hope that the correction will meet with approval. Once again, thank you very much for your comments and suggestions.

Round 2

Reviewer 2 Report

Comments and Suggestions for Authors

The authors have already address my comments successfully, and accept is suggested.

Comments on the Quality of English Language

The current version is ok, but can be improved a bit.

Author Response

Dear Reviewer,

Thank you for your comments concerning our manuscript entitled “A charging planning method for shared electric vehicles with the collaboration of mobile and fixed facilities” (Manuscript ID: sustainability-2699150). Your comments are very helpful for improving our manuscript.

We have made corresponding changes for better editing of English language. Revisions are marked in red, such that any changes can be easily viewed.

We sincerely hope that this revised manuscript would meet with your approval. We appreciate for your warm work earnestly.